



# Detecting long-term changes in point source fossil $CO_2$ emissions with tree ring archives

**E. D. Keller[1], J. C. Turnbull[1,2] and M. W. Norris[1]**

[1]{National Isotope Centre, GNS Science, Lower Hutt, New Zealand}

[2]{CIRES, University of Colorado at Boulder, CO, USA}

Correspondence to: E. D. Keller (l.keller@gns.cri.nz)

**Abstract**

We examine the utility of tree ring $^{14}C$ archives for detecting long term changes in fossil $CO_2$ emissions from a point source. Trees assimilate carbon from the atmosphere during photosynthesis, in the process faithfully recording the average atmospheric $^{14}C$ content in each new annual tree ring. Using $^{14}C$ as a proxy for fossil $CO_2$, we examine interannual variability over six years of fossil $CO_2$ observations between 2004-05 and 2011-12 from two trees growing near the Kapuni Natural Gas Plant in rural Taranaki, New Zealand. We quantify the amount of variability that can be attributed to transport and meteorology by simulating constant point source fossil $CO_2$ emissions over the observation period with the atmospheric transport model WindTrax. We compare model simulation results to observations and calculate the amount of change in emissions that we can detect with new observations over annual or multi-year time periods given both measurement uncertainty of 1ppm and the modelled variation in transport. In particular, we ask, what is the minimum amount of change in emissions that we can detect using this method, given a reference period of six years? We find that changes of 42% or more could be detected in a new sample from one year at the same observation location, or 22% in the case of four years of new samples. This threshold lowers and the method becomes more practical the more the size of the signal increases. For point sources 10 times larger than the Kapuni plant (a more typical size for power plants worldwide), it would be possible to detect sustained emissions changes on the order of 10% given suitable meteorology and observations.



## 1 Introduction

Carbon dioxide ($CO_2$) emitted by anthropogenic activity is the largest single contributor to the radiative forcing causing climate change (IPCC, 2014). It thus plays a crucial role in any attempt to prevent or mitigate further warming. Large point sources (mainly from electricity generation and industry) contribute around a third of the total fossil-fuel derived $CO_2$ ($CO_2$ff) emissions (IPCC, 2014) and in many places are included in government regulatory schemes that aim to reduce emissions (e.g. European Union ETS, South Korea, Switzerland, and others at the city/state level; Serre et al., 2015). Emissions are typically reported on an annual basis, and reduction targets are commonly agreed as annual or multi-year caps, often requiring changes in emissions relative to a baseline year (e.g. the Kyoto Protocol and the new Intended Nationally Determined Contributions (INDC), UNFCCC, 2015).

Emissions are currently known from "bottom up" techniques such as self-reported data from fuel usage statistics (Boden et al., 2015) and/or continuous stack monitoring (U.S. Environmental Protection Agency, 2005; eGRID, 2014) and are subject to significant uncertainties (Ackerman and Sundquist, 2008; Gurney et al., 2009, 2012). This uncertainty might include not only methodological biases and possible deliberate underreporting but also simple error in compiling statistics. The integrity of regulation schemes and their effectiveness at limiting future climate change will require independent methods of evaluating reported emissions and improvement in the accuracy of emissions inventories (Tans and Wallace, 1999; Nisbet and Weiss, 2010; National Research Council, 2010; Gurney, 2013).

"Top-down" atmospheric observations can provide an independent method for evaluating emissions. This involves taking observations of atmospheric gas mole fractions in combination with atmospheric transport modelling to infer the magnitude of emissions from a source or region over a particular time period (e.g. McKain et al., 2012; Lindenmaier et al., 2014; Brioude et al., 2013). It can be quite challenging to quantify absolute values of emissions and $CO_2$ fluxes in general because of the large errors and biases typically encountered in transport models (e.g. Stephens et al., 2007; Lin and Gerbig, 2005; Gerbig et al., 2008; Prather et al., 2008; Geels et al., 2007; Liu et al., 2011; Kretschmer et al., 2012). However, *relative* changes in emissions are usually easier to determine, since any consistent biases in the model will cancel out. By establishing a baseline measurement over a reference period, we can compare future observations to this reference and calculate relative changes



that occur. In this manner, we can potentially verify relative emission reduction targets
without requiring precise knowledge of the absolute levels of emissions.
One of the biggest challenges of atmospheric observations of $CO_2$ff is distinguishing the
fossil component from the considerable background level of $CO_2$ that occurs naturally in the
atmosphere, currently about 400 parts per million (ppm; Mauna Loa observation record,
http://www.esrl.noaa.gov/gmd/ccgg/trends/index.html, last access: 13 May 2015). In addition,
there are large diurnally and seasonally varying $CO_2$ fluxes from the biosphere, which may
result in changes in $CO_2$ mole fraction of tens of ppm within a single day at near-surface sites
(e.g. Miles et al., 2012). This problem can be avoided by using the [14]C isotopic content as a
tracer for $CO_2$ff. $CO_2$ff contains no [14]C: the half-life of [14]C is 5,730 years (Karlen et al.,
1968), and all of the [14]C has decayed away from fossil fuels. Other sources of $CO_2$ have
roughly the same [14]C content as the atmosphere. By measuring the [14]C content of $CO_2$ or a
proxy for $CO_2$, we can calculate the portion of observed $CO_2$ that comes from recently added
fossil fuel emissions (Levin et al., 2003; Meijer et al., 1996; Turnbull et al., 2006).
Plant material can be used as a proxy for atmospheric $CO_2$ff because plants assimilate carbon
from the atmosphere during photosynthesis, in the process faithfully recording the [14]C content
in new plant material. The radiocarbon content in tree rings has been well established as a
tracer for fossil $CO_2$ emissions (Suess, 1955; Tans et al., 1979; Djuricin et al., 2012;
Rakowski et al., 2013) and as a method to detect leaks from $CO_2$ geosequestration (Donders
et al., 2013). Tree rings represent an integrated average of daytime $CO_2$ atmospheric mole
fractions and [14]C content over the tree's annual growth period. This allows for a retroactive
analysis of $CO_2$ff mole fractions over many years, including any trends in emissions that
occurred during the life of the tree.
In this study, we evaluate whether we can detect changes in $CO_2$ff emission rates from a point
source on an annual time scale using the $CO_2$ff mole fraction derived from the [14]C content of
tree ring archives. Variations in the observed $CO_2$ff mole fraction at a given location are
dependent on not only the emission rate but also on atmospheric transport, which in turn is
subject to naturally varying meteorological conditions (e.g. wind speed and direction,
temperature, pressure, etc.). Detecting a change in the emission rate requires disentangling
this change from the natural variability in transport and meteorology as well as from
measurement uncertainty in the observations. The question we ask in this paper is: can we use
tree ring archives to detect changes in $CO_2$ff emissions from a point source, and if so, what is





the minimum change in annual emissions that we can detect given the typical measurement
uncertainty of 1ppm and natural variability in transport? A similar analysis was carried out by
Levin and Rodenbeck (2007) at the regional scale, using a 20-year time series of $^{14}$C
observations over Germany. McKain et al. (2012) also assessed the ability of an observation-
model framework to detect changes in regional urban $CO_2$ emissions on a monthly time scale.
We re-examine this question on the scale of an individual point source with mean annual
observations.
We calculate interannual variability in observations from tree ring archives of annual
(growing season) $CO_2$ff between 2004-05 and 2011-12[1], taken from two different trees
growing south of the Kapuni Natural Gas Treatment Plant in rural New Zealand (Norris,
2015). We then use an atmospheric transport model, WindTrax, with local meteorological
data to quantify the interannual variability that can be expected due to measurement
uncertainty, transport and meteorology at different distances and orientations from the source,
including the locations of the trees. Finally, we look at what this implies for detection limits in
the context of emissions monitoring or verification and practical considerations in the
presence of multiple sources of uncertainty.

## 18    2    Methods

### 19    2.1    Site

The site of our study is the Kapuni Natural Gas Treatment Plant in rural Taranaki, New
Zealand (39.477° S, 174.1725° E, 170 m.a.s.l.) (Fig. 1). This site was chosen because it is
located in flat terrain and is relatively isolated from other sources of $CO_2$ff, considerably
simplifying measurement and analysis. The gas treatment plant, owned and operated by
Vector, processes natural gas extracted from natural gas wells in the Taranaki Basin. The gas
contains around 40% $CO_2$, which is removed during processing and vented to the atmosphere
at a rate of ~0.1 TgC yr$^{-1}$ (NZMED, 2010). In addition, there is an ammonia urea
manufacturing plant 500m to the west of the gas plant (Fig. 1), operated by Ballance Agri-
Nutrients, which also releases $CO_2$ff to the atmosphere during the manufacturing process.
This site emits roughly a third of the amount of the Vector gas plant (~0.03 TgC yr$^{-1}$)

---

[1] Henceforth in this paper, the growing season spanning 1 September to 30 April will be referred to by the year
in which the season began, i.e. 2004-05 will be designated 2004.




(Taranaki Regional Council, 2013). Although the signal from the Vector plant is much
stronger, especially to the east (downwind from the dominant westerly winds), emissions
from the Ballance plant are potentially large enough to detect at some locations and are
included in our simulations unless otherwise specified.
The surrounding terrain is flat and mostly free of obstructions, with elevation varying no more
than 10m within 2km of the plant. The largest nearby topographic feature is a dip of ~5m into
the Kapuni stream immediately east of the Vector emission source. The landscape is
dominated by highly productive pasture grazed by dairy cows, with large and diurnally
varying $CO_2$ fluxes. The prevailing wind direction is from the west, with a smaller proportion
from the southeast and north (Figs. 2 and 3).
## 2.2  $CO_2$ emissions
Emissions data were supplied by Vector as monthly totals (Peter Stephenson, personal
communication), which we have converted to average daily rates for the purpose of
modelling. Mean annual daily emissions for each year between 2004 and 2011 from 1
September to 30 April are shown in Fig. 4; data are listed in Table S1. The long-term mean is
5341 gC s$^{-1}$, with a standard deviation in annual means of 388 (7.3%). There are annual
fluctuations but no long-term trend over the modelled period 2004-2011. The largest change
during a single year occurred in 2008, when the emissions dropped by 14% relative to the
mean. On a longer time scale, there are more significant changes, including the start of
operations at the Vector Plant in 1971. However, we focus on the 2004-2011 period during
which high resolution local meteorological data is available. There are no significant seasonal
or diurnal variations in the emissions of which we are aware.
The Ballance Agri-Nutrients Plant emissions are reported on an annual basis (Taranaki
Regional Council, 2013). Average daily rates in each growing season are depicted in Fig. 4.
The mean daily rate of emissions over the period 2004-2011 is 1512 gC s$^{-1}$ with a standard
deviation in annual means of 88 (18%), which is more variable than the emissions from the
Vector plant, but smaller in absolute terms. Emissions vary somewhat from day to day
according to production levels, but more detailed daily or monthly information is unavailable;
for simplicity we assume a constant emissions rate in each year. We note that emissions are
much lower in 2011, which is due to downtime after both a fire and scheduled maintenance
(Taranaki Regional Council, 2013).





## 2.3 Tree ring observations

Tree rings faithfully record the $^{14}C$ content of assimilated $CO_2$, so when the rings are independently dated by dendrochronology, we can determine an average $^{14}C$ content and recently added $CO_2$ff in the local atmosphere for the period during which the tree ring was laid down. We use core samples from two trees located south of the plant, a pine tree (*Pinus radiata*) and a chestnut tree (*Castanea sativa*) (Fig. 1; Norris, 2015). The pine tree is located in a stand of trees within 5m of the Kapuni stream, with the crown reaching 10m above the associated terrain dip. The chestnut is isolated in a flat paddock.

Each tree ring is assumed to represent the Southern Hemisphere summer growth period from 1 September to 30 April, as this is when the majority of plant photosynthesis occurs and new plant material is produced. The sample preparation, measurement and determination of $CO_2$ff are described in detail by Norris (2015). In summary, alpha cellulose was extracted from individual rings, combusted, reduced to graphite and measured by accelerator mass spectrometry. $CO_2$ff was determined following Turnbull et al. (2014) from the isotopic difference between the measured tree ring and clean air background $CO_2$ measured at Baring Head, Wellington (41.4167°S, 174.8667°E; Currie et al., 2011; extended dataset to 2015 will be presented in an upcoming publication). Baring Head, located at the southern end of New Zealand's North Island and approximately 220 km southeast of Kapuni, was chosen as the background for this study over more local sites because it provides a long-term record of background $CO_2$ and $^{14}C$, dating back to the early 1970s. Background levels in tree rings measured at a site in Kapuni 2km upwind of the Vector plant are close to those measured at Baring Head in the same time period, justifying the use of the Baring Head dataset (Norris, 2015). Uncertainty in $CO_2$ff is dominated by $\Delta^{14}C$ measurement uncertainty in both background and the observed sample and is typically ~1ppm for this dataset.

The process of $CO_2$ adsorption in plants is extremely complex. For simplicity, we assume a constant assimilation rate over all daylight hours. In reality, $CO_2$ adsorption varies with plant species and photosynthesis rates, being weighted towards sunny periods and midday (Bozhinova et al., 2013). There are also many different climatic and nutrient limitations that can only be properly accounted for with a full process-based biogeochemical model of plant growth, which is beyond the scope of this study. We do, however, take into consideration the fact that plant material will tend to underestimate mean $CO_2$ff when $CO_2$ff is variable, as in the case of a plume from a point source (see Sect. 2.7).





## 2.4 WindTrax model

WindTrax (WindTrax 2.0; Thunder Beach Scientific, Nanaimo, Canada, www.thunderbeachscientific.com) is a Lagrangian particle dispersion model used to estimate unknown trace gas concentrations or emission rates from a source over short distances (~1km). WindTrax has been applied to agricultural emissions from area sources, such as methane, ammonia, and other gasses from grazing dairy cows, cattle feedlots and farm waste (e.g. Flesch et al., 2005; Laubach and Kelliher, 2005; Bonifacio et al., 2013; Rhoades et al., 2010; Wilson et al., 2012; McBain and Desjardins, 2005). It has also been assessed in the context of $CO_2$ sequestration leakage detection (Leuning et al., 2008; Loh et al., 2009). Modelling integrated averages of $CO_2$ff in plant material is a relatively new application. WindTrax was chosen for this study because it is easy to use and the distance scale is appropriate for our site. We previously used WindTrax to estimate $CO_2$ff in grass samples at the Kapuni site (Turnbull et al., 2014), demonstrating that the model is capable of providing reasonable estimates of observed $CO_2$ff. Here, we take the same approach to model $CO_2$ff measured in tree rings. We note that WindTrax is not applicable to complex terrain or larger distance scales and caution is urged when applying our methodology to other sites.

WindTrax simulates the transport of trace gases by releasing a set number of particles at each time step and following each particle's trajectory downwind. Based on Monin–Obukhov similarity theory, the physics underlying the model is described in detail in Flesch et al. (2004) and Wilson and Sawford (1996). The model equations are valid in the atmospheric surface layer. It assumes wind and other meteorological observations are averaged over a suitable time interval representing a stable, mean atmospheric state (10-30 minute intervals are recommended). Intervals longer than one hour have been shown to be problematic (Flesch et al., 2004) because at these time intervals, large-scale fluctuations not built in to the model become important. In this study, we use one hour time steps to match the resolution of our meteorological dataset (see Sect. 2.5).

The model can be run in forward (fLS) or inverse/backward (bLS) mode, depending on whether the emissions or the trace gas mole fractions are unknown. In all simulations described here we start with known emission rates and use the fLS mode to estimate the $CO_2$ff mole fraction at locations surrounding the plant. Model "concentration sensors" represent simulated measurements of mole fractions at designated locations and supply the main model output.





The model is stochastic, meaning that it introduces random turbulence into particle
trajectories, and no two runs are identical, even with the same parameters and meteorological
input. There is, therefore, inherent error in the model predictions due to the randomness
introduced in the transport process. Only the average behaviour of a group of particles can be
determined, and releasing more particles at each time step will tend to reduce the degree of
uncertainty. Statistical error (or the standard deviation within each set of trajectories) is
calculated and output by the model at each time step. However, any biases in the modelled
transport or the meteorological input data used to drive the model are not accounted for.
**2.5  Meteorology**
Modelling with WindTrax requires at a minimum wind speed, wind direction, air temperature,
and atmospheric pressure at each time step. We use hourly meteorological data from the
Hawera Automatic Weather Station (AWS) (39.6117°S, 174.2917°E, 98 m.a.s.l), downloaded
from the New Zealand National Climate Database (CliFlo, 2014). Hawera, approximately
20km distance to the southwest of Kapuni, is the nearest location with a nearly complete long-
term dataset of hourly wind direction and speed. Eight years of data (2004-2011) were
available at the time of our study. We use only data from the growing season (1 September –
30 April) and daylight hours (08:00 – 18:00 local daylight savings time) in the model
simulations to correspond to the time period during which trees assimilate $CO_2$.
The area to the northwest of Hawera and Kapuni is dominated by Mount Taranaki, a 2518m
volcanic cone that rises steeply from relatively flat surrounding terrain. Wind direction and
speed can be very different at sites only a few kilometres apart because of the local impact of
the mountain on atmospheric flow. Thus we compared Hawera and Kapuni meteorological
datasets to ensure that Hawera is representative of Kapuni over long (~1 year) time periods
and the wind speed and direction distributions as a whole are similar at both locations. A wind
rose for the eight years (2004-2011) of data at Hawera is shown in Fig. 2, together with a
wind rose for one year (2013) of data at Kapuni. Wind speeds are on average higher at
Hawera, but the distribution in direction is very similar, with a small overrepresentation of
northerlies at Hawera. The wind speed and direction distributions at both locations are shown
in more detail in Fig. S1.
We demonstrate correlation between the two sites using the only overlapping dataset that was
available for direct comparison at the time of the study. We collected data at a temporary





meteorological station at Kapuni at 10-minute intervals during the period 14 August – 26
October 2012, with some significant data gaps (Turnbull et al., 2014). These were averaged to
hourly intervals and compared with the corresponding set of measurements at the Hawera
AWS. Only daylight hours were included for consistency with the model simulations. Using
these datasets, correlation in wind speed is good, with $R^2$ = 0.82, and correlation in wind
direction is moderate ($R^2$ = 0.61). Because wind direction is an angular measurement,
correlation in wind direction was performed using the circular package v0.4-7 in R v3.0.2
(Lund and Agostinelli, 2013; R Core Team, 2013) rather than the standard linear correlation
function. Scatter plots comparing wind at Kapuni and Hawera directly at each time step are in
Fig. S2. Wind speed is a good match, with Hawera on average having slightly higher speeds
than Kapuni.  With wind direction, most points are close to the 1:1 line or slightly below,
indicating a small rotation in direction between the sites. Approximately 67% of data points
(one sigma) are within 30° of each other, and 85% are within 45°. For the purpose of our
simulation in which we focus on integrated averages rather than particular points in time, the
Hawera dataset is sufficiently representative of typical conditions at Kapuni.
We expect variability in $CO_2$ff mole fraction to be strongly related to variability in wind
speed and direction, and consequently sampling location. Annual mean wind speed does not
vary by much; the mean wind speed over all eight years is 6.3 m s$^{-1}$, and the standard
deviation in annual mean is 0.11 m s$^{-1}$, which is only 2% of the mean. Mean wind direction is
273° (from the west), but there is also a significant amount of wind from the southeast and
north-northeast (Figs. 2 and 3). This general pattern did not change from year to year over the
eight years of the simulation, but relative proportions in each direction did sometimes vary
considerably (Fig. 3). In particular, northerlies (the direction most relevant to our
observations) range from 21-28% of the total, a 30% change in the northerly fraction. While
always the largest category, the percentage of westerlies varies between 38-52%. It is notable
that there are very few periods with calm winds; the region is in general very windy.
**2.6   Model parameters**
Several model parameters are held constant throughout all simulations. The modelled surface
is short grass (surface roughness $z_o$ = 2.3cm), since the majority of the surrounding area is
grazed dairy pasture. The heights of the two emissions stacks are set to their known values:
35m above ground level for Vector and 36m for Ballance. The model's atmospheric stability
parameter is also held constant using the general class of 'moderately unstable'. While this is



not true for all modelled time periods, in the absence of measurements from a 3D sonic
anemometer or other reliable indicators of atmospheric stability, a general stability class is a
first approximation. We tested the model at a different constant stability class ('slightly
unstable') and found no significant difference in the amount of variability (results not shown).
We note, however, that atmospheric stability is a potential source of error; others have found
that stability is an important parameter that can bias results, and model estimates are generally
improved with input from a sonic anemometer or vertical profiles of wind speed and
temperature (Flesch et al., 2004; Gao et al., 2009; Koehn et al., 2013).
Model concentration sensors at the locations of the pine and chestnut trees are placed at
heights of 15.0m and 5.0m, respectively, reflecting the approximate height of the canopy. A
single height at each tree was chosen to reduce model complexity and runtime; however, we
recognize that in reality $CO_2$ is assimilated over a range of heights at each tree, corresponding
to the vertical spread of the canopy. Some previous studies have indicated that concentrations
modelled with WindTrax are sensitive to sampling height and/or the ratio of sampling height
to distance from the source (e.g. McBain and Desjardins, 2005; Laubach and Kelliher, 2005;
Laubach, 2010). To test for dependence on height, we simulated $CO_2$ff along a 20m vertical
profile at the location of the pine and chestnut trees (results not shown). Results vary
somewhat according to height, and averaging over a 5m height range slightly reduces the
mean and interannual standard deviation, but not enough to change our results significantly.
**2.7  Simulations**
We ran a "constant emissions, variable meteorology" simulation at an hourly time step with
all eight years of available meteorological data from Hawera (excluding night time and winter
months), concentration sensors placed at the locations of the trees, and both the Vector and
Ballance plants as $CO_2$ff point sources (Fig. 1). Because emissions are held constant, this
simulation enables us to isolate contributions to variability from meteorology and transport.
For each tree, four concentration sensors were placed on the vertices of a square, with sides of
length 30m, centred on the location of the trees and averaged to reduce model transport error.
The emission rate at each source was the reported mean rate over the entire modelled period.
In addition to the model sensors at the locations of the trees, we placed sensors at hypothetical
locations in four directions and two horizontal distances from the emissions source to
examine more general model sensitivity and variability due to meteorological conditions at





our site without being tied to the locations of specific observations. Eight additional sensors
were placed 1.5m above the ground in the four cardinal directions relative to the Vector plant,
one each at 300m and 600m horizontal distance from the source. Only one point source, the
Vector plant, was included in the results at these sensors to simplify analysis. Emissions are
constant at the Vector mean rate over the eight years.
We also ran a "constant meteorology, variable emissions" simulation in which we repeat the
meteorology from one year (2004) and allow emissions rates to vary according to the reported
values. This allows us to examine model annual variability due to emissions, independent of
transport.
We subsequently generated a "variable emissions, variable meteorology" simulation by
scaling modelled mole fractions at the tree rings from the constant emissions, variable
meteorology simulation according to reported emissions levels in each year (Fig. 4). This is
valid because the relationship between source strength and concentration flux passing through
a location downwind is linear (Leuning et al., 2008). In addition, under unstable atmospheric
conditions the emissions leave the model domain within one hour and do not return, so data in
a given year is not affected by the emissions from previous years. This simulation is used to
compare the model to observations.
Because plant material will underestimate mean $CO_2ff$ when $CO_2ff$ is variable, rather than
comparing the tree ring measurements to the raw model output of $CO_2$ mole fractions, we
calculate a modelled "$CO_2ff_{tree}$". This is the $CO_2ff$ that the model would predict from the plant
material given measured background levels and the equations governing $\Delta^{14}C$. We use the
following equations:
$$\Delta_i = \frac{\Delta_{bg} C_{bg} + \Delta_{ff} C_{ff\,i}}{C_{bg} + C_{ff\,i}} \tag{1}$$
$$\Delta_{tree} = \frac{1}{N} \sum_{i=1}^{N} \Delta_i \tag{2}$$
$$C_{ff\,tree} = \frac{C_{bg}(\Delta_{tree} - \Delta_{bg})}{\Delta_{ff} - \Delta_{tree}} \tag{3}$$
where $\Delta = \Delta^{14}C$, $C_{ff\,i}$ is the modelled $CO_2ff$ at the $i^{th}$ time step, $N$ is the total number of model
time time steps, $C_{bg}$ and $\Delta_{bg}$ are measured (Norris, 2015), and $\Delta_{ff} = -1000$. The basic
derivation of this equation can be found in Turnbull et al. (2006). This accounts for the fact





that plant material will assimilate roughly the same amount of $CO_2$ at each time step
regardless of the variability in atmospheric $CO_2$ mole fraction induced by the emission plume,
and thus the $\Delta^{14}C$ of the plant material represents a simple mean of the $\Delta^{14}C$ in the assimilated
$CO_2$ at each time step. In contrast, sampling of whole air across the same time period would
collect more $CO_2$ during times of high $CO_2$ mole fraction, weighting the resultant $\Delta^{14}C$
towards these periods. This results in a $CO_2ff_{tree}$ that is lower than would be obtained by
determining the simple mean $CO_2ff$ from the modelled mole fractions. Model results from the
variable emissions simulation reported in Fig. 4 and Sect. 3 were derived using these
equations.

## 3    Results and Discussion

### 3.1    Observation and model comparison

We first compare modelled $CO_2ff_{tree}$ to the observed tree ring $CO_2ff$ to evaluate the model's
ability to estimate annual integrated averages in this context and to identify possible biases
and error in the model. Our observations from tree rings consist of six annual measurements
of $CO_2ff$ from both the pine tree and the chestnut tree between 2004 and 2011 (2008 and 2010
are missing) (Fig. 4). The means over this period are 5.4ppm (pine) and 2.1ppm (chestnut)
(Table 1). Mean modelled $CO_2ff_{tree}$ over the same six years (excluding the two years without
observations, 2008 and 2010) is 6.1ppm and 2.2ppm for the pine and chestnut tree,
respectively. The modelled mean is almost an exact match for the chestnut tree (difference of
0.1ppm) and within error for the pine tree (difference of 0.7ppm). Figure 4 shows a direct
comparison between measured and modelled $CO_2ff$ for each year. At the pine tree, model
performance is very good: four of the six (66%) annual observed values are within one sigma
of the modelled values, and the remaining two are within two sigma. The agreement for
individual years at the chestnut tree is poorer, but with large errors in the observations and the
distance from the source close to the limit of model capabilities, this is expected.
The model is able to simulate both the long-term mean and the annual variation in $CO_2ff_{tree}$
with a reasonable degree of accuracy, and there are no significant biases apparent. Thus we
can be confident that the model is representative of relative interannual variability in
transport, which is the focus for the remainder of this paper.





## 3.2 Drivers of interannual variability in $CO_2$ff

Detecting changes in emissions requires disentangling the changes in $CO_2$ff due to emissions from other sources of interannual variability. We now examine the variability in our observations and turn to our model simulations to determine the relative contributions from emissions, transport, and measurement uncertainty.

The observed standard deviations of the six annual $CO_2$ff values from the tree rings are 0.8ppm (14% of the six-year mean) and 1.1ppm (51%) for the pine and chestnut tree, respectively (Table 1). This includes not only variability in emissions but other sources of uncertainty such as meteorology and transport, variable $^{14}$C assimilation rates in the trees, precision of measurements, and background corrections. Measurement uncertainty in particular is important at these relatively small concentrations. Given that the standard deviations are very close to the typical measurement uncertainty of ~1ppm, the scatter in annual means can be attributed in large part to this factor alone. For example, at the pine tree, we would expect at least four out of six measurements to be within 1ppm (one sigma) of the long-term mean, all other factors being constant. This is indeed true of four of the six observations. Measurement uncertainty is proportionally much higher in the case of the chestnut tree, which is ~1km from the Vector plant and where the average signal is only ~2ppm. At this distance measurement uncertainty would seemingly dominate other sources of variability. In contrast, the pine tree is much closer to the source (~400m), and the signal is two to three times larger. Variations in emissions will make up a larger proportion of the total variation and are more likely to be detectable at current measurement precision.

The standard deviations of modelled $CO_2$ff$_{tree}$ in the variable emissions, variable meteorology simulation are 0.5ppm (7.8%) and 0.3ppm (15%) at the pine and chestnut tree, respectively (Table 1). Adding measurement uncertainty of 1ppm in quadrature, we would predict the standard deviations of the annual means in observed $CO_2$ff to be 1.1ppm (18%) and 1.0ppm (47%) for the pine and chestnut, respectively, if variability in emissions, atmospheric transport and measurement uncertainty explain all of the interannual variability. In comparison, the observed standard deviations of the annual means are 14% of the long-term mean at the pine tree and 51% at the chestnut tree. Thus emissions, transport, and measurement uncertainty are able to explain the interannual variability in the observations within error.





We can estimate the relative proportion of interannual variability that is due to atmospheric
transport using the constant emissions model simulation, in which the only source of
variability is meteorology. The modelled mean $CO_2$ff over the six years with observations is
7.4ppm and 2.7ppm for the pine and chestnut, respectively, and modelled standard deviations
are both 0.5ppm (6.6% and 19% of the respective means) (Table 1). Over the full eight years
of the model simulation, the means and standard deviations are 7.7 / 0.9 ppm (12%) and 2.7 /
0.5 ppm (19%), respectively.
Examining more general patterns of meteorological and transport variability at the Kapuni site
apart from the locations of the trees reveals that the variation is highly dependent on the
direction of the observation location relative to the source. The results at the eight
hypothetical sensors averaged in each individual year and means for the entire eight years of
simulation are compared in Fig. 5, and the long-term means and standard deviations are given
in Table 2. The variation to the south of the plant (10-11% of the mean) is the lowest of any
direction and consistent with the variation found at the pine tree in the constant emissions
simulation over the full eight years (12%). Absolute $CO_2$ff mole fractions are highest in the
east (westerlies being dominant), but standard deviations are slightly higher at 14% of the
mean. Concentrations in the west are low (~2ppm) and highly variable, the result of the low
percentage of easterlies in any given year (Fig. 3). Variation is relatively insensitive to the
distance from the source.
It is apparent that wind direction drives a large part of the variation in transport. Annual
modelled $CO_2$ff at the trees in the constant emissions simulation is correlated with the annual
percentage of wind in the direction +/- 30° of the direct line between the source and the tree,
corresponding to the plume trajectories that are most likely to pass through the tree locations
(Fig. S3; $R^2$ = 0.56 and 0.72 for the pine and chestnut tree, respectively). The same correlation
between wind direction and modelled $CO_2$ff at all eight hypothetical sensors combined gives
an $R^2$ of 0.58. Over half of the transport variability is thus explained solely by variation in the
percentage of wind in each direction. However, other meteorological variables and model
parameters (e.g. wind speed, temperature, pressure, etc.) still play a non-negligible role, as the
annual variation in wind direction is not equivalent to the interannual variability in modelled
$CO_2$ff.
In the same manner, we can determine the contribution of changes in emission rates to the
overall interannual variability with the constant meteorology simulation in which emissions


vary but transport is the same in each year. This results in interannual variability in $CO_2$ff
similar to the variability in the emissions themselves, with the magnitude roughly scaled to
the distance from the emission source: the standard deviations are 0.5ppm (7.4%) and 0.2ppm
(7.6%) for the pine and chestnut tree, respectively. In comparison, the standard deviation of
the average daily emissions rate over the six years with observations is 7.9% of the mean for
the Vector plant and 21% for the Ballance plant, with a standard deviation of 8.1% for the
combined total (over the full eight years between 2004 and 2011, the standard deviations are
7.3% and 18% of the 8-year mean for Vector and Ballance emissions, respectively, and the
variation in the combined emissions is 7.7%). This is on the same order of magnitude of the
variability due to transport at the pine tree but only about half the amount at the chestnut tree.
Looking at all of the factors together (Table 1), variations in emissions and transport
contribute about equally to total variation at the pine tree. At the chestnut tree, transport
makes up a larger proportion of the total, which likely reflects the greater variability in
meteorology in that particular direction. The variability in emissions somewhat counter-
balances the variability in transport, particularly at the chestnut tree, where the standard
deviation with both variable emissions and meteorology (0.3ppm / 15%) is lower than that
with constant emissions (0.5ppm / 19%). This is most likely coincidental to the particular
years of observations, as there is no correlation between variations in emissions and variations
in transport (not shown). Meteorological variation happens to be lowest in the south, where
the trees are located, even though the largest signal occurs to the east (Table 2 and Fig. 5). In
this respect, the trees are fortuitously located for our study. This underscores the benefit of
analysing transport variability at a particular location before collecting observations, as the
quality of results can be greatly influenced by meteorological patterns.
**3.3   Detection limits**
Given the amount of interannual variation in meteorology and transport that we can infer from
the model and typical measurement uncertainty of 1ppm, what is the minimum change in
emissions that it is possible to detect in a tree ring sample taken at Kapuni, representing an
integrated average of $CO_2$ff over a year or more? We use a student t-test to quantify the
minimum amount of change in observations required (relative to the long-term average or
reference period) that would allow us to conclude that there has been a change in emissions.
The t-test calculates the minimum difference between the long-term mean and a new annual
tree ring sample (or samples) that would be statistically significant above scatter or noise from





other factors. We make the assumption that our observations and simulated mole fractions are
normally distributed. The results of the 2-sided test (representing change in either direction) at
a 95% confidence level are given in Table 3 for "future" samples representing one, two and
four years of integrated average $CO_2$ff. All percentages are relative to the long-term mean
over six years, our reference period for this study. We assume that the standard deviation in
future samples due to interannual variability in meteorology is the same as the standard
deviation over the reference period.
Using the modelled means and standard deviations from the constant emissions simulation of
tree ring $CO_2$ff and measurement uncertainty of 1.0ppm, the detection limits represent the
minimum observed change that would indicate a driver of variability other than transport or
measurement uncertainty, in this case $CO_2$ff emissions. With a new observation representing
one year (i.e. one tree ring), the difference between the long-term mean and the new sample
would need to be more than 42% at the pine tree and 115% at the chestnut tree to have high
confidence that the sample shows a change in emissions, rather than just natural variability or
uncertainty. If we have four new annual observations at the new emission rate, the difference
reduces by half to 22% and 62%, respectively. These detection thresholds are well above the
reported annual changes in emission rates between 2004 and 2011 (Fig. 4). At the distance
and location of the chestnut tree (~1km), it seems likely that the signal is too small and
variable to be practical for detecting emission changes for a point source with emissions of
this magnitude.
If we relax the condition to one sigma (or a 68% confidence level), would we be able to detect
the largest change in emissions reported at the Vector Plant between 2004 and 2011? The
student t-test at 68% confidence level gives corresponding detection limits listed in Table 3.
For a one-year observation from the pine tree, this is 18%. The largest change in emissions in
any single year at the Vector plant is in 2007, with a decline of 14% relative to the long-term
mean, still below the detection limit. Indeed, looking at the results in Fig. 4, there is no
significant decline at the chestnut tree in 2007; there is a small decline in $CO_2$ff at the pine
tree but it is too small to conclude that emissions have changed. If we were able to achieve a
reduction in measurement uncertainty to 0.5ppm, however, the threshold for detection at the
pine tree becomes an 11% change in emissions, and we would expect to be able to observe a
14% decline in emissions. In this case, the small decline in $CO_2$ff at the pine tree in 2007
would be significant.



Would we be able to detect this change at a different location (in direction and/or distance)
around the Kapuni plant? Our hypothetical concentration sensors 300m and 600m from the
source (Table 2) indicate that with a single one-year $CO_2$ff observation, only a change in
emissions of at least 36% would be detectable at 95% confidence, a much larger change than
occurs in our observational dataset. The location of the pine tree (at 400m southeast of the
plant) appears to provide as good a detection capability as any of our hypothetical sensors.
However, if we have four years of observations (and the change in emissions was sustained
over that time period) located either to the east or the south of the plant at a distance of 300m,
we would be able to detect a change of 10% or more at the one-sigma confidence level.
Changes of 20% or more would be detectable at these same locations with one year of
observations, or alternately, four years of observations if we require high confidence.
This analysis uses the actual meteorology only to determine the interannual variability in
$CO_2$ff that we might expect due to meteorological variations.  If we also know the
meteorology in the year or years of the new observations, we can quantify the change in
emissions by modelling transport at constant emissions. For example, attributing 15% of the
one-year variation at the pine tree to the combined factors of transport and measurement
uncertainty (Table 1) and assuming that the rest of the variation is due to emissions, this
translates to a change in emissions of 27% over the one year. In this manner it is possible to
get a more precise estimate of the long-term changes in emissions.
Additionally, if we have multiple measurements over the same period at different locations
around the point source, measurement uncertainty reduces proportionally by $1/\sqrt{n}$, where $n$ is
the number of independent measurements. The resulting reduction in detection thresholds is
more complex and depends on the long-term mean and variation at each of the observation
locations. One could, for example, use a paired t-test to determine if the change detected in all
of the measurements taken together is significant. This is beyond the scope of the current
study, but the detection thresholds given in Tables 2 and 3, based on a single observation
location, would overestimate the minimum change in emissions that it is possible to observe
with multiple measurements designed to cover the area surrounding the point source.
**3.4  Applicability to other point sources**
The results presented here are specific to the meteorology and point sources at the Kapuni
site, but the methodology can be extended to any point source with suitable trees growing





nearby. Ideally, observations would be made as close to the source as possible in the direction
where the signal is strongest and/or most consistent. If measurement uncertainty of 1ppm is to
be relatively unimportant compared to the combined transport and emissions variability of 8%
at the pine tree (i.e. adding measurement uncertainty does not change the total variation in
measured $CO_2$ff by more than 1-2%), we require a signal around 20-30ppm, implying a
required emission rate five times that of the Kapuni Vector plant. Alternatively, if we were
able to reduce measurement uncertainty to 0.5ppm (for example, by increased measurement
precision or taking measurements from multiple locations at the site), we would be able to
detect changes with signals at around half the magnitude, and the method could be more
feasible for emission sources the size of the Kapuni Vector Plant. Additionally, if we have
multiple measurements from the same period at various locations surrounding the source,
detection thresholds lower further and we can achieve the same sensitivity with a smaller
point source.
Our case study involves point sources that are fairly small on an international scale; for
comparison, the world's largest power plant, in Taiwan, emits about 300,000 gC s$^{-1}$ or 9.5
TgC yr$^{-1}$ (Ummel, 2012), which is 95 times as much as the Vector plant at Kapuni.  There are
approximately 800 power plants worldwide that emit more than 10 times the annual total
$CO_2$ff at Kapuni (CARMAv3.0, 2009; Wheeler and Ummel, 2008; Ummel, 2012). The
typical emission rates seen at these larger power plants would produce signals in which
measurement uncertainty is only a small proportion of the total. With annual signals
theoretically 10 times that observed at the Kapuni pine tree and the same amount of
meteorological variation, all other things being equal, the detection threshold for a one-year
measurement at the location of the pine tree would be 19%, or 10% with four years of
measurements. This is a plausible reduction target, and the method would be useful for
verifying emissions changes in such cases.

## 4    Conclusions

We have examined sources of interannual variability in $CO_2$ff in samples from tree ring
archives representing integrated averages over one year. We used the atmospheric transport
model WindTrax to separate variability in meteorology and transport from other sources of
variation in our observations. At the location of the pine tree, modelled variation in transport
is 7% of the six-year reference mean. This is about the same magnitude as the variation in




emissions that were recorded over the same time period. At the chestnut tree, variation due to
atmospheric transport is larger, at 19% of the mean, and is about twice the magnitude of the
variation in emissions. Taking into account typical measurement uncertainty of 1ppm for
radiocarbon samples, in order to conclude with high confidence that there has been a change
in emissions and not just natural variation in meteorology, we would require an observed
change of 42% from the mean in a new one-year sample from the pine tree. If we take a two-
year or four-year sample average, this reduces to 30% and 22%, respectively. This is well
above the largest single-year change in emissions at the Vector Plant, which is 14%.
However, if we are able to reduce measurement uncertainty by half, to 0.5ppm, or if the point
source doubles in strength, detection thresholds are closer to the actual level of variation in
emissions. If we only require confidence at the one-sigma level, we would in this case be able
to detect a 14% change in a single year.
Detection limits are highly dependent on the location of the observations and specific
meteorology of the site. Wind patterns should be carefully considered before deciding where
to take samples in any study, preferably in an area where the signal will be strongest and
where wind patterns will be most consistent through time. A model analysis such as we have
performed can give an idea of the baseline variability in transport and the size of the signal
needed to observe changes in emissions. This makes it theoretically possible to separate the
uncertainty in transport from other sources of uncertainty.
In general, this method will be most effective when observations are made in the dominant
wind direction and/or in a direction with consistent winds, close enough to the point source so
that natural variability in meteorological conditions and measurement uncertainty does not
overwhelm the signal from the emissions. The larger the point source (the higher the emission
rate) and the signal from $CO_2$ff, the more able integrated averages from plant material will be
to detect changes in emissions. For larger power plants or other point sources of a more
typical size worldwide, detecting changes with this method could be feasible; with signals 10
times or more the size of Kapuni, measurement uncertainty is relatively insignificant, and
sustained changes in emissions on the order of 10% can be detected from a single sampling
location given suitable meteorological conditions and observations.
**Acknowledgements**



This work was funded by GNS Science Strategic Development Fund and public research
funding from the Government of New Zealand. Marcus Trimble assisted with sample
collection, dendrochronology and sample preparation. The Rafter Radiocarbon Laboratory
staff (Jenny Dahl, Johannes Kaiser, Kelly Lyons, Christine Prior, Helen Zhang and Albert
Zondervan) were invaluable in their assistance with the radiocarbon measurements. We thank
Peter Stephenson and the staff at the Vector Kapuni Gas Processing Plant for providing
information on the plant's $CO_2$ emissions and allowing us to sample trees on their site. We
also thank Roger Luscombe for allowing access to his land and chestnut tree for sampling and
his extensive knowledge of local history.





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





Table 1. Observed and modelled $CO_2ff$ means and standard deviations. All means and
standard deviations (SD) include six years (2008 and 2010 are omitted because there are no
observations available for these years). Measurement uncertainty (MU) of 1.0ppm is
explicitly added to the modelled results in the far right column. Observations implicitly
include this uncertainty.

| Observation or simulation (2004-2011) | Mean (ppm) | SD (% of mean) | SD + 1.0ppm MU (% of mean) |
|---|---|---|---|
| **Pine** | | | |
| Observed | 5.4 | | 0.8 (14%) |
| Modelled $CO_2ff_{tree}$ : variable meteorology, variable emissions | 6.1 | 0.5 (7.8%) | 1.1 (18%) |
| Modelled $CO_2ff$ : variable meteorology, constant emissions | 7.4 | 0.5 (6.6%) | 1.1 (15%) |
| Modelled $CO_2ff$ : constant meteorology, variable emissions | 7.3 | 0.5 (7.4%) | 1.1 (15%) |
| **Chestnut** | | | |
| Observed | 2.1 | | 1.1 (51%) |
| Modelled $CO_2ff_{tree}$ : variable meteorology, variable emissions | 2.2 | 0.3 (15%) | 1.0 (47%) |
| Modelled $CO_2ff$ : variable meteorology, constant emissions | 2.7 | 0.5 (19%) | 1.1 (41%) |
| Modelled $CO_2ff$ : constant meteorology, variable emissions | 2.3 | 0.2 (7.6%) | 1.0 (43%) |



Table 2. Eight-year modelled mean CO$_2$ff and standard deviation (SD) of eight hypothetical
sensors for eight years of constant emissions simulation and detection limits at the two-sigma
(95%) and one-sigma (68%) confidence level (CL) for samples representing an average of
one, two, or four years.

| Model Sensor | Mean (ppm) | SD (% of mean) | SD + 1ppm MU (% of mean) | % change detectable (95% CL) | | | % change detectable (68% CL) | | |
|---|---|---|---|---|---|---|---|---|---|
| | | | | 1 yr | 2 yr | 4 yr | 1 yr | 2 yr | 4 yr |
| North 300m | 12.2 | 2.4 (20%) | 2.6 (21%) | 53% | 38% | 29% | 24% | 18% | 13% |
| North 600m | 4.6 | 0.8 (18%) | 1.3 (29%) | 72% | 52% | 39% | 33% | 24% | 18% |
| East 300m | 22.8 | 3.2 (14%) | 3.3 (15%) | 37% | 27% | 20% | 17% | 12% | 9.4% |
| East 600m | 9.0 | 1.3 (14%) | 1.6 (18%) | 45% | 33% | 24% | 20% | 15% | 12% |
| South 300m | 11.7 | 1.3 (11%) | 1.7 (14%) | 36% | 26% | 20% | 16% | 12% | 9.2% |
| South 600m | 4.7 | 0.5 (10%) | 1.1 (24%) | 60% | 43% | 33% | 27% | 20% | 15% |
| West 300m | 1.6 | 0.8 (50%) | 1.3 (81%) | 204% | 148% | 111% | 92% | 68% | 52% |
| West 600m | 0.34 | 0.16 (50%) | 1.0 (300%) | 744% | 540% | 405% | 337% | 250% | 190% |



Table 3. Detection limits for samples at trees, calculated with modelled $CO_2$ff at constant
emissions and six years of observations in reference period. Limits are given at the two-sigma
(95%) and one-sigma (68%) confidence level (CL) for samples representing an average of
one, two, or four years. Measurement uncertainty (MU) of 1.0ppm or 0.5ppm is added in
quadrature to the standard deviation of modelled $CO_2$ff before limits are calculated.

| Modelled $CO_2$ff: variable meteorology constant emissions | % change detectable (95% CL) | | | % change detectable (68% CL) | | |
|---|---|---|---|---|---|---|
| | 1 yr | 2 yr | 4yr | 1 yr | 2 yr | 4yr |
| **Pine** | | | | | | |
| Modelled $CO_2$ff + 1.0 MU | 42% | 30% | 22% | 18% | 13% | 10% |
| Modelled $CO_2$ff + 0.5 MU | 27% | 19% | 14% | 11% | 8.5% | 6.5% |
| **Chestnut** | | | | | | |
| Modelled $CO_2$ff + 1.0 MU | 115% | 83% | 62% | 92% | 68% | 52% |
| Modelled $CO_2$ff + 0.5 MU | 89% | 64% | 48% | 38% | 28% | 22% |





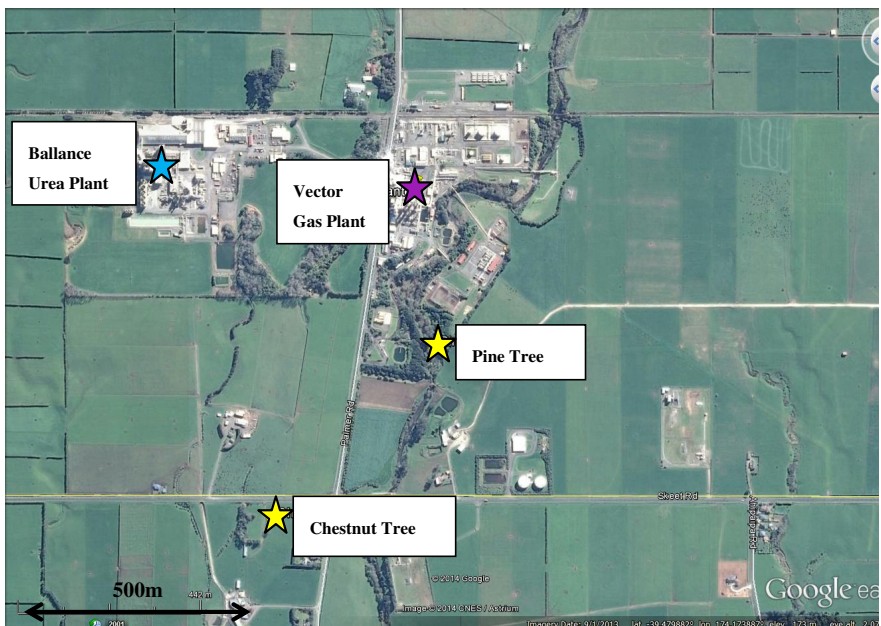

3   Figure 1. Aerial view of Kapuni area, with the sampled pine and chestnut trees and Vector

4   Gas Treatment Plant and Ballance Agri-Nutrient Urea Plant labelled.




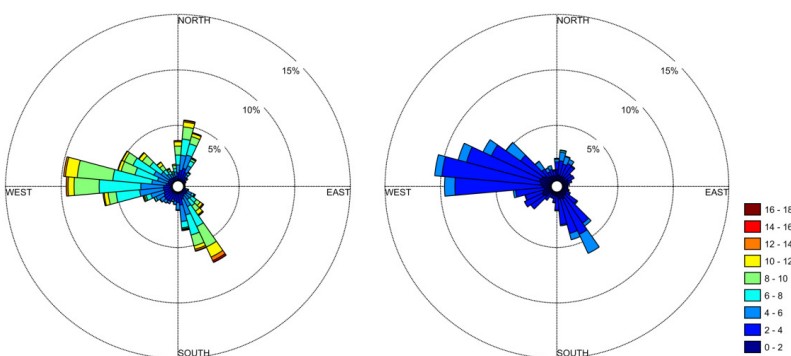

3    Figure 2. Wind roses during the growing season at Hawera 2004-2011 (left) and Kapuni 2013

4    (right), daylight hours only (8:00am – 6:00pm).





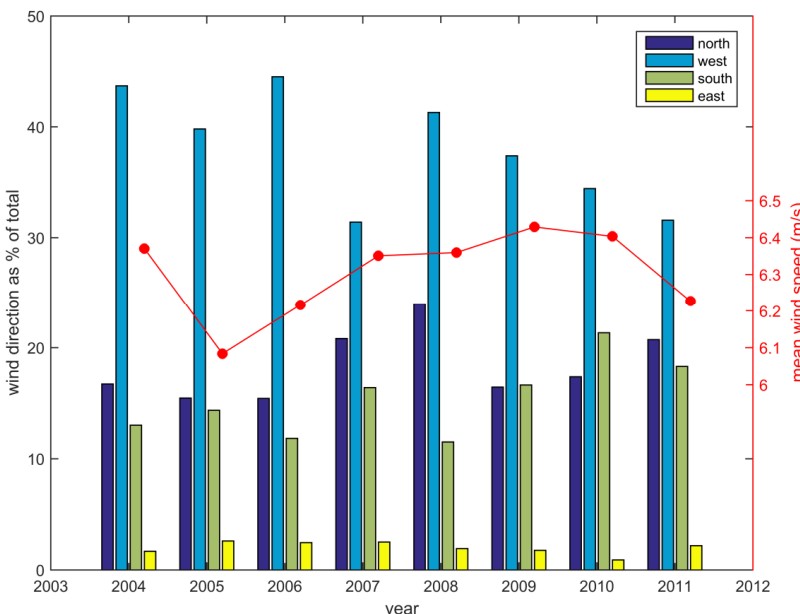

Figure 3. Percentage of wind in each of four directions (left axis) and mean wind speed (right
axis) by growing year between 2004 and 2011 (daylight hours only, 8:00am – 6:00pm).
Directions are defined by +/- 30 degrees due north, west, south, and east (i.e. west is defined
as wind from 240° to 300°). Note that this does not comprise the complete 360° circle so
percentages do not add to 100.



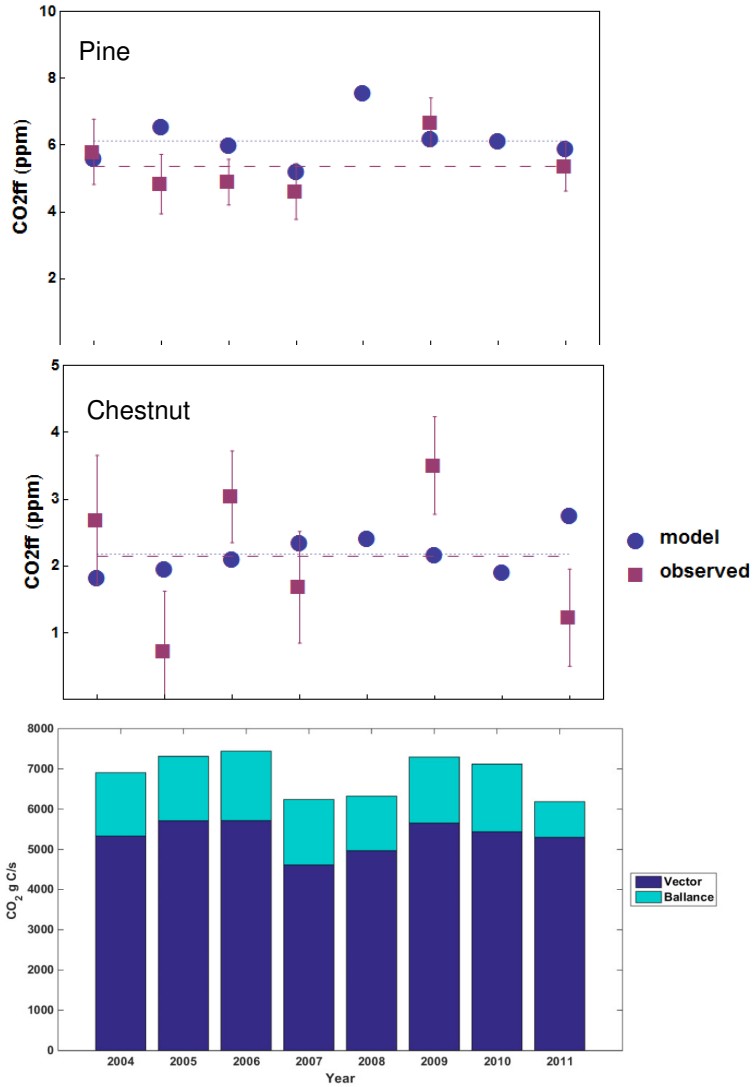

3    Figure 4. Pine tree (top) and chestnut tree (middle) modelled $CO_2ff_{tree}$ vs. tree ring observed

4    $CO_2ff$. Dashed lines show modelled and observed six-year means. Bottom panel shows the

5    average emissions rate for Vector and Ballance in each year.





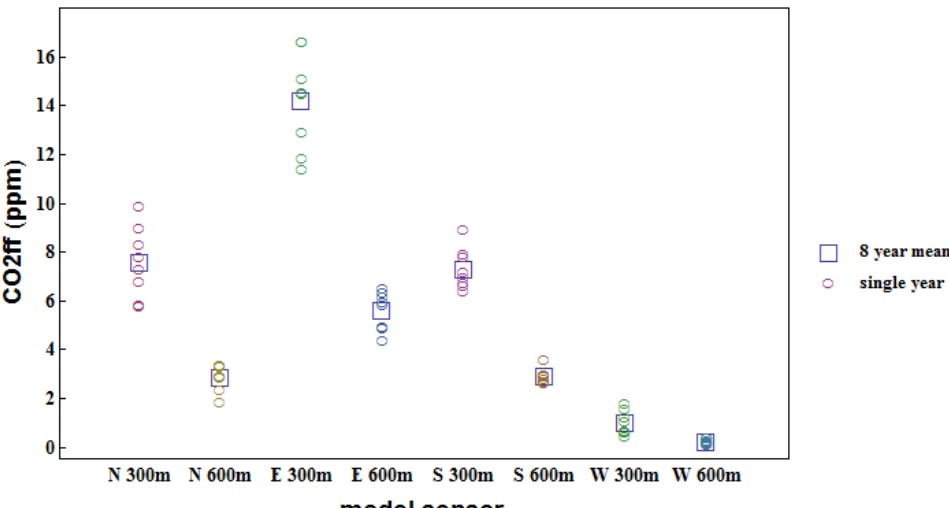

Figure 5. Constant emissions, variable meteorology simulation results for hypothetical
sensors: $CO_2$ff mole fraction averaged over all eight years of simulation (squares) and
individual annual averages (circles). Sensors are labelled by direction (N, E, S or W) and
distance (300m or 600m) from the source.