# Peer review of "Detecting long-term changes in point source fossil CO2"

_Atmospheric Chemistry and Physics, 2015_

## Referee Comment (RC1) · Anonymous Referee #1 · 10 Feb 2016

In this study tree ring 14C archives were used to detect long-term changes in fossil CO2 emissions from point sources at Taranaki, New Zealand. The atmospheric transport model WindTrax was used to simulate constant point source fossil CO2 emissions over the observation period to quantify the amount of variability that can be attributed to transport and meteorology. Model simulation results were compared to observations and the minimum detectable amount of change in emissions over a one-year period, two years, and four years of sampling was investigated. Determining changes that occur in CO2 emissions relative to a baseline measurement is very useful, as it can facilitate the verification of relative emission reduction targets without a precise knowledge of the absolute emission level.

Overall the manuscript is well structured and fits the scope of ACP. I recommend some revisions before this manuscript can be published.
[Figure]

1. The authors describe the site to be located in a flat terrain dominated by Mount Taranaki. They also point out that the wind direction and speed can be very different at sites only a few kilometers apart due to the mountain influence on atmospheric flow. Hawera is considered to be representative of Kapuni in the manuscript, but comparisons of wind speed and direction were made for a very short time interval "14 August-26 October 2012, with some significant data gaps" as specified by the authors (page 9, line 1). I doubt that two months of measurements (with significant gaps) are sufficient to consider the two locations similar from this prospect. Also, the wind speed at Hawera is approximately double compared to the wind speed at Kapuni (not "slightly higher speeds" as mentioned on page 9, line 10). I wonder how much does this affect the model results? It is not clear to me what data is used in Fig. 2 for the wind rose "Kapuni 2013".

2. It is not clear from the text how many trees were sampled from each species. How many replicates were used and how was the data analyzed? Please specify and expand this paragraph (page 6, lines 9-24) rather than pointing to papers only. The reader should have a clear idea about the tree ring observation methodology without reading Norris, 2015 and Turnbull et al, 2014.

3. When describing the model, the authors state that this is appropriate for estimating emission rates from a source over short distances (page 7, line 4). They also show that the time interval recommended for the meteorological observations used for the model is 10-30 min. How reliable are the model results for these simulations given that one hour time-step was used for wind speed/direction? Also, the chestnut tree is located at the limit of the simulation capability, 1 km. How does this influence the result?

4. The authors present this method to be useful for verifying emission changes at other locations where the point sources are much stronger, mentioning that there are approximately 800 power plants worldwide that emit more than 10 times the annual total CO2ff at Kapuni (page 18, line 17). They also explain that "WindTrax is not applicable to complex terrain or larger distance scales and caution is urged when applying our

methodology to other sites". I have a feeling that Kapuni site is very specific and I am not sure that there are so many other sites with flat terrain, trees within 300-600m of the point source located downwind, and consistent winds through time. What other model would then be most suitable for complex terrain and larger distances? Add suggestions for other model(s) that would be suitable in this case.

Specific comments:

Check the table captions. Information is missing (e.g. Table 2 – column 4 not explained).

Check figure captions. The reader should understand what those figures represent without reading the text.

Figure 2: name the two panels a) and b) and refer to them in text accordingly. Expand the caption.

Figure 4: Same as for Fig. 2.

Page 5, line 18: 2008 should be 2007.

Page 16, lines 26-28: "Indeed, looking at the results in Fig. 4, there is no significant decline at the chestnut tree in 2007; there is a small decline in CO2ff at the pine tree but it is too small to conclude that emissions have changed. " As I estimate from the figure, the observed value is smaller in 2007 than in 2006 at the chestnut tree by ~1.3 ppm, and by ~0.3 ppm at the pine tree. Isn't the former significant? I recommend using the same scale for the two graphs.

---

## Referee Comment (RC2) · Anonymous Referee #2 · 15 Feb 2016

This is an excellent paper showing the use of radiocarbon analysis of tree rings to document changes in fossil fuel CO2 emissions on a small spatial scale. It describes a study using two trees close to a natural gas processing plant and a urea processing plant in New Zealand, far removed from other major anthropogenic sources of CO2. Radiocarbon measurements of annual tree rings are used as proxies for CO2ff and compared with the known emissions of the natural gas processing plant, using results from atmospheric transport modeling by WindTrax. The modeling results are used to evaluate the sources of uncertainty and to determine what magnitude variations in emissions could be detected from point sources using this method. The magnitude of detectable inter-annual variations depends heavily on the magnitude of the emissions source. For a small source, such as that studied here, the change in emissions from one year to the next must be 42%, 22% for four years of new samples. However,

for a more typical source, such as a major power plant, a 10% sustained change in emissions would be detectable, if the sampling location is appropriate in terms of wind direction and speed. This method could be very important for monitoring emissions reductions from major point sources.

Overall this paper is very well written and the arguments are generally easy to follow. It is appropriate for publication in Atmospheric Chemistry and Physics with some revisions.

Comments: I have questions about the meteorology used for the modeling. You compare the limited data set at Kapuni, close to the sampling site, with the much more complete set at Hawera, 20 km southwest of the sampling site. You state that the correlations between wind speed and wind direction between the two sites are consistent enough to warrant using the complete Hawera data set, as shown in a direction comparison for limited dates during August-October 2012 (Fig. S2). But is the limited period in 2012 adequate for evaluating whether Hawera data are appropriate for modeling wind transport at the Kapuni site? Moreover, Figures 2 and S1 show that the wind speed at Hawera (6-7 m/s) averages on the order of twice that at Kapuni (2-3 m/s). Have you done any sensitivity calculations to see how this difference in wind speed affects the modeling? The wind directions seem to be fairly consistent at the two sites.

p.1, line 25: change "lowers" to "is reduced"

p.2, line 9: rearrange "reduction targets are commonly agreed as" to "commonly agreed upon reduction targets are"

p. 3, lines 16-17: You mention here and again later "the [photosynthesis] process faithfully recording the 14C content in new plant material", but you only reference the work showing this significantly after the mention on p. 11. It might help the reader to have this discussion earlier, since it is critical to the method.

p. 5, line 18: "2008" should probably be "2007".

p. 6, line 13: insert $\Delta$14C before "measured".

p. 6, lines 15-20: How long are the more local sites' records? Is it important that the background site have data back to the early 1970s if the data you are looking at only starts in 2004?

p. 6, line 32: Add "end of" before "Sect. 2.7".

p. 9, line 18: Add "at Hawera" after "eight years".

p. 11, lines 14-16: Is this also true for the moderately unstable conditions used here?

p. 11, line 18: Add "as explained below" after "variable".

p. 11, line 23: Is there any Cbio (local biosphere contribution)?

p. 11, line 27: Since background is very important to these calculations and the thesis describing these background samples is not readily available, can you describe what samples were used for background and how they were determined to be appropriate background samples?

p. 12, lines 1-7: Can you move this to earlier in the discussion, perhaps above the equations?

p. 14, line 7: You need a conclusion statement explicitly relating these numbers to meteorology.

p. 14, lines 13-14: The hypothetical sensor at W 600 m has smallest variability.

p. 14, line 19: Add "(Table 2)" at the end of the sentence.

p. 18, Conclusions: Add context to the names of the trees, since some people might read only abstract and conclusions! Examples below.

p. 18, line 31: After "pine tree", add ", 400 m from the major source".

p. 19, line 1: After "chestnut tree", add "1 km from the major source".

Figures: In general, increase font sizes for labels. Label panels within figures "a", "b", "c" to make it easier to refer to them in the text.

Figure 1: Can you add a large-scale location map locating Taranaki in New Zealand, as well as Hawera and Mount Taranaki? Add a label for Kapuni stream.

Figure 2: Font sizes. Label the legend (m/s).

Figure 4: The bottom axis of the top panel is missing. Increase the font size of the axis tick labels in all panels. The dates don't line up between the top two panels and the bottom panel. Increase all font sizes for the bottom panel. You use a subscript for $CO_2$ in the bottom panel, but not in the top two. In the caption: "Dotted and dashed lines show modeled and observed six-year means, respectively."

Figure 5: What do the different colors for the circles indicate? The legend only shows the purple color.

---

## Author Response (AR1)

"Detecting long-term changes in point source fossil CO2 emissions with tree ring 2 archives"

E.D. Keller et al.

**Response to Referee Comments**

Thank you to both referees for your comments. Each comment is copied below in red, and our
response appears immediately after in black. Changes made to the text appear in italicised
blue.

**9 Comments from Anonymous Referee #1:**

1. The authors describe the site to be located in a flat terrain dominated by Mount Taranaki.
They also point out that the wind direction and speed can be very different at sites only a few kilometers apart due to the mountain influence on atmospheric flow. Hawera is considered to be representative of Kapuni in the manuscript, but comparisons of wind speed and direction were made for a very short time interval "14 August-26 October 2012, with some significant data gaps" as specified by the authors (page 9, line 1). I doubt that two months of measurements (with significant gaps) are sufficient to consider the two locations similar from this prospect. Also, the wind speed at Hawera is approximately double compared to the wind speed at Kapuni (not "slightly higher speeds" as mentioned on page 9, line 10). I wonder how much does this affect the model results? It is not clear to me what data is used in Fig. 2 for the wind rose "Kapuni 2013".

Our response also addresses the following comment from Anonymous Referee #2:

Comments: I have questions about the meteorology used for the modeling. You compare the limited data set at Kapuni, close to the sampling site, with the much more complete set at
Hawera, 20 km southwest of the sampling site. You state that the correlations between wind speed and wind direction between the two sites are consistent enough to warrant using the complete Hawera data set, as shown in a direction comparison for limited dates during

August-October 2012 (Fig. S2). But is the limited period in 2012 adequate for evaluating whether Hawera data are appropriate for modelling wind transport at the Kapuni site?

Moreover, Figures 2 and S1 show that the wind speed at Hawera (6-7 m/s) averages on the order of twice that at Kapuni (2-3 m/s). Have you done any sensitivity calculations to see how

**this difference in wind speed affects the modeling? The wind directions seem to be fairly consistent at the two sites.**

Thank you to both referees for raising this issue. We have re-examined all of the wind data 4 that is available to us from Kapuni, and in particular the data from 2013-14 that is shown in 5 Fig. 2 and Fig. S1. This data came from a temporary automated weather station (AWS) that was installed in Sep 2013 at the Shell Todd Oil Services (STOS) gas production station, 6 7 adjacent to the north side of the Vector Gas Plant. It was hired from and maintained by a third 8 party and removed in Dec 2014, and we unfortunately do not have any record of its 9 calibration or maintenance. We compared several other independent datasets (the 10-minute 10 wind speed and direction from the temporary weather station installed by the authors between 11 14 Aug - 26 Oct 2012 that is already mentioned in the manuscript, daily mean wind speeds 12 available through New Zealand's Virtual Climate Station Network (VCSN; Tait et al., 2006), 13 10-minute wind speed from an AWS installed by Vector on-site at the Kapuni Gas Plant covering Aug 2012, Oct 2012, Nov 2013, and Sep-Dec 2014, and two weeks of measurements 14 15 from a sonic anemometer installed by the authors in a nearby paddock at Kapuni in Oct 16 2014), in which the mean wind speeds at Kapuni are on average only 80-90% lower than 17 those at Hawera and, where there is overlap, are in disagreement with the wind speeds in the 18 STOS dataset. Consequently, we believe that the wind speeds from the STOS AWS are biased 19 low, and the true relationship between the wind speeds at Hawera and Kapuni is close to that 20 derived from our data measured between Aug and Oct 2012. (We are confident of the quality 21 of this data because we installed the weather station and verified the instrument calibration 22 ourselves.) The wind direction is for the most part consistent in all data sources.

The low wind speeds measured at the STOS AWS could be due to either poor instrument 24 calibration or the placement of the station itself. We emphasize that we did not use the STOS 25 dataset in our modelling, but only for general comparison with the data from Hawera. 26 Because we now doubt the accuracy of the wind speeds from the STOS dataset, we have 27 replaced the data in Fig. 2b with our dataset from Aug-Oct 2012, and also added a wind rose 28 from Hawera during the same time period for direct comparison. As more evidence of the 29 similarity of Kapuni and Hawera, we have added a comparison of daily mean wind speeds 30 from the VCSN, and have included a histogram from this dataset as Fig. S1. The text has been 31 edited to remove all mention of the STOS dataset. We acknowledge that the correlation is still 32 based on a very limited dataset, and that this is a potential source of error in our results. 1 However, we have no reason to think that these months were atypical of conditions at Kapuni; other months and seasons for which we have data follow the same general patterns.

We did perform a sensitivity test of the effect of wind speed on the modelled results. There is 4 an inverse proportional relationship between wind speed and modelled concentrations, so that 5 halving the wind speed approximately doubles the modelled CO2ff concentrations. If the wind 6 speeds at Kapuni are significantly lower than those at Hawera, our results would be an under-7 prediction. The statement "slightly higher speeds" to which Referee #1 referred is based on 8 model II linear regression performed on the overlapping dataset between 14 Aug - 26 Oct 9 2012. This equation is y = 0.90x - 0.32, where x is wind speed at Hawera and y is wind speed 10 at Kapuni (calculated using R package Imodel2 major axis regression). This equation is now 11 mentioned in the text and the caption of Fig. S2. The additional data sources that we have 12 examined show a similar relationship, and we maintain that wind speed and direction at 13 Hawera is similar enough to Kapuni for this study, which does not depend on exact point-by-14 point correlation. The revised text reads (p. 9 line 8 - p. 10 line 11): 15 The area to the northwest of Hawera and Kapuni is dominated by Mount Taranaki, a 2518m volcanic cone that rises steeply from relatively flat surrounding terrain. Wind direction and 16 17 speed can be very different at sites only a few kilometres apart because of the local impact of the mountain on atmospheric flow. Thus we compared Hawera and Kapuni meteorological 18 19 datasets to ensure that Hawera is representative of Kapuni over long (~1 year) time periods 20 and the wind speed and direction distributions as a whole are similar at both locations. A 21 wind rose for the eight years (2004 2011) of data at Hawera is shown in Fig. 2, together with a wind rose for one year (2013) of data at Kapuni. Daily mean wind speeds were compared 22 using the Virtual Climate Station Network (VCSN; Tait et al., 2006). This is a set of "virtual" 23 24 weather stations that uses re-analysis interpolation techniques to provide historical daily weather variables on a 5 x 5 km grid across New Zealand. The mean wind speed at Hawera 25 over the modelled time period, 5.0 m  $s^{-1}$ , is only slightly higher than that at Kapuni, 4.6 m  $s^{-1}$ . 26 Histograms comparing the wind speed distributions at both sites are in Fig. S1. Wind speeds 27 are on average higher at Hawera, but the distribution in direction is very similar, with a 28 29 small overrepresentation of northerlies at Hawera. The wind speed and direction

- 30 distributions at both locations are shown in more detail in Fig. S1.
- 31 We demonstrate correlation between the two sites using the only Only one overlapping
- 32 dataset with sub-daily time intervals that was available for direct comparison at the time of the our study. We collected data at a temporary meteorological station situated in a paddock 2 at Kapuni at 10-minute intervals during the period 14 August – 26 October 2012, with some significant data gaps (Turnbull et al., 2014). These were averaged to hourly intervals and 3 4 compared with the corresponding set of measurements at the Hawera AWS. Only daylight 5 hours were included for consistency with the model simulations. Wind roses for the Kapuni 6 dataset and the corresponding time period at Hawera are shown in Figs. 2b and 2c. The 7 distribution in direction is similar to the north, but there are more southerlies and fewer 8 westerlies at Hawera. Using these datasets, correlation in wind speed is good, with  $R^2 = 0.82$ , and correlation in wind direction is moderate ( $R^2 = 0.61$ ). Because wind direction is an 9 angular measurement, correlation in wind direction was performed using the circular 10 package v0.4-7 in R v3.0.2 (Lund and Agostinelli, 2013; R Core Team, 2013) rather than the 11 12 standard linear correlation function. Scatter plots comparing wind speed and direction at Kapuni and Hawera directly at each time step are in Fig. S2. Wind speed is a good match, 13 14 with Hawera on average having slightly higher speeds than Kapuni. When wind speed at Hawera is linearly regressed against wind speed at Kapuni, the resulting equation is y =15 16 0.90x - 0.32. (Model II regression was performed with the Imodel2 v1.7-2 package in R 17 v3.0.2 (Legendre, 2014)). With wind direction, most points are close to the 1:1 line or slightly 18 below, indicating a small rotation in direction between the sites. Approximately 67% of data 19 points (one sigma) are within  $30^{\circ}$  of each other, and 85% are within  $45^{\circ}$ . For the purpose of 20 our simulation in which we focus on integrated averages rather than particular points in time, 21 the Hawera dataset is sufficiently representative of typical conditions at Kapuni. We note, 22 however, that the dataset from Kapuni spans a very limited time period, and this is a potential 23 source of error in our results. 24 References added:

Tait, A., Henderson, R., Turner, R., and Zheng, X. G.: Thin plate smoothing splineinterpolation of daily rainfall for New Zealand using a climatological rainfall surface, Int. J.

Climatol., 26, 2097-2115, 2006.

Revised Figure 2:

~

Figure 2. Wind roses at hourly intervals a) at Hawera during the eight growing season (SepApr) between 2004-2011, b) Hawera 14 Aug – 26 Oct 2012, and c) Kapuni 14 Aug – 26 Oct
2012, all showing daylight hours only (8:00am – 6:00pm). Wind speed is in m s-1. Data at
Kapuni was collected at 10-minute intervals and averaged to hourly intervals to match
Hawera data.

Revised Figure S1:

Figure S1. Histograms of daily mean wind speeds (m s-1) at Hawera (a) and Kapuni (b) for
the eight growing seasons 2004-2011 from the VCSN. Dashed red line shows the mean over
the entire period (5.0 and 4.6 m s-1 for Hawera and Kapuni, respectively).

2. It is not clear from the text how many trees were sampled from each species. How many
replicates were used and how was the data analyzed? Please specify and expand this
paragraph (page 6, lines 9-24) rather than pointing to papers only. The reader should have a
clear idea about the tree ring observation methodology without reading Norris, 2015 and
Turnbull et al, 2014.

Additional details about the tree ring measurements have been added. The equation in 12 Turnbull et al., 2014 used to derive  $CO_2$ ff from the measurements has also been added. The 13 revised text reads (p. 6 line 12 – p.7 line 10):

In summary, wood was sampled from the trees using a Haglöff incremental borer. Four cores

- 15 were extracted per tree at equidistant points at a height of approximately 1.2m from the base
- 16 of the tree. One core from each tree was used to create a historic record of CO2 emissions from commission of the Kapuni plant in 1971 to the outermost ring at the time of sampling in

- 2 2012. Replicates were taken from a second core to validate ring counting and  ${}^{14}C$  results.
- 3 Alpha cellulose was extracted from individual rings using a method modified from Hua et al.
- 4 (2000), combusted with a Europa ANCA elemental analyser (EA), reduced to graphite and
- 5 measured by accelerator mass spectrometry at GNS Science laboratories in Lower Hutt, New

Zealand (Baisden et al., 2013; Zondervan et al., 2015; Turnbull et al., 2015).

- 7 CO2ff was determined following Turnbull et al. (2014) from the isotopic difference between
- 8 the measured tree ring and clean air background CO2 measured at Baring Head, Wellington
(41.4167°S, 174.8667°E; Currie et al., 2011; extended with unpublished data dataset to 2015
- 10 will be presented in an upcoming publication). Baring Head, located at the southern end of
- 11 New Zealand's North Island and approximately 300 km south of Kapuni, was chosen as the
- 12 background for this study over more local sites because it provides a long-term record of
- 13 background  $CO_2$  and 14C, dating back to the early 1970s. The following equation was used:

$$C_{ff} = \frac{C_{obs} \left( \Delta_{obs} - \Delta_{bg} \right)}{\left( \Delta_{ff} - \Delta_{bg} \right)} - \beta$$
(1)

where  $C_{\rm ff}$  is CO2ff,  $C_{obs}$  is the CO2 mole fraction in the observed sample,  $\Delta_{obs}$  and  $\Delta_{bg}$  are the 15  $\Delta^{14}C$  of the observed sample and background sample, respectively.  $\Delta_{\rm ff}$  is the  $\Delta^{14}C$  of CO2ff, 16 and is assigned to be -1000%.  $\Delta_{be}$  is from the summer season average from the long-term 17 Wellington 14CO2 record at Baring Head. Comparison of this record with tree rings collected 18 3 km upwind of our source showed no difference from the Wellington record.  $\beta$  is a small 19 correction to account for the fact that the  $\Delta^{14}C$  of  $CO_2$  from other sources may be slightly 20 21 different from that of the atmosphere; in our case we set  $\beta$  to zero since the proximity to the 22 coast and consistent winds suggest that  $CO_2$  other is negligible in this location (Turnbull et 23 al., 2014). Baring Head, located at the southern end of New Zealand's North Island and approximately 220 km southeast of Kapuni, was chosen as the background for this study over 24 25 more local sites because it provides a long term record of background CO2 and 14C, dating 26 back to the early 1970s. Background levels in tree rings measured at a site in Kapuni 2km 27 upwind of the Vector plant are close to those measured at Baring Head in the same time period, justifying the use of the Baring Head dataset (Norris, 2015). Uncertainty in CO2ff is 28 dominated by  $\Delta^{14}C$  measurement uncertainty in both background and the observed sample 29 30 and is typically ~1ppm for this dataset.

References added:

- Baisden,W. T., Prior, C. A., Chambers, D., Canessa, S., Phillips, A., Bertrand, C., Zondervan,
   A., and Turnbull, J. C.: Radiocarbon sample preparation and data flow at Rafter:
   accommodating enhanced throughput and precision, Nucl. Instrum. Meth. B, 294, 194–198,
   2013.
- Hua, Q., Barbetti, M., Jacobsen, G.E., Zoppi, U. and Lawson, E.M.: Bomb radiocarbon in
  annual tree rings from Thailand and Australia, Nucl. Instrum. Meth. B, 172, 359-365, 2000.
- 7 Zondervan, A., Hauser, T.M., Kaiser, J., Kitchen, R.L., Turnbull, J.C. and West, J.G.:
- 8 XCAMS: The compact 14 C accelerator mass spectrometer extended for 10 Be and 26 Al at
- 9 GNS Science, New Zealand, Nucl. Instrum. Meth. B, 361, 25-33, 2015.
- 10

3. When describing the model, the authors state that this is appropriate for estimating emission rates from a source over short distances (page 7, line 4). They also show that the time interval recommended for the meteorological observations used for the model is 10-30 min. How reliable are the model results for these simulations given that one hour time-step was used for wind speed/direction?

This information comes from Flesch et al., 2004, which provides detailed analysis of the 17 effect of averaging time on WindTrax model results. The model is built on a traditional Monin-Obukhov similarity theory (MOST) description of the atmosphere and relationships 18 19 derived from 15-60 min wind statistics. WindTrax is limited by the fact that large-scale 20 atmospheric dispersion fluctuations are not incorporated in the model structure. As the time 21 interval increases, large-scale motions become more important, and Flesch et al., 2004 states 22 that applying the model to "time averaging periods greatly different from 15-60 min carries a 23 risk" and an increase in error. While the preferred choice of time step is given as 10-30 24 minutes, 60-min time intervals are still in the range considered valid for application of MOST 25 statistics. We believe that using a one-hour time step in this context does not make the model 26 unreliable. We have edited the text to clarify this point and more accurately reflect the 27 language in the original reference (p. 8 lines 7-12):

It assumes wind and other meteorological observations are averaged over a suitable time
interval representing a stable, mean atmospheric state (model relationships are built from
wind statistics over 15-60 minute intervals; 10-30 minute intervals are recommended using
model time steps greatly outside of this range is not recommended). Intervals longer than one hour have been shown to can be problematic (Flesch et al., 2004) because at these time
 intervals, large-scale fluctuations not described by MOST statistics become important.

Also, the chestnut tree is located at the limit of the simulation capability, 1 km. How does this
influence the result?

The referee correctly points out that the chestnut tree is located at the limit of WindTrax's 7 capability. We discuss this in sections 3.1 and 3.3, attributing the large errors and high 8 detection thresholds at least in part to the tree's distance from the point source. The small 9 concentrations combined with the large model error make this distance impractical for 10 detecting changes in  $CO_2$  emissions. We have not made any changes to the text.

4. The authors present this method to be useful for verifying emission changes at other 13 locations where the point sources are much stronger, mentioning that there are approximately 800 power plants worldwide that emit more than 10 times the annual total 14 15 CO2ff at Kapuni (page 18, line 17). They also explain that "WindTrax is not applicable to 16 complex terrain or larger distance scales and caution is urged when applying our 17 methodology to other sites". I have a feeling that Kapuni site is very specific and I am not 18 sure that there are so many other sites with flat terrain, trees within 300-600m of the point 19 source located downwind, and consistent winds through time. What other model would then 20 be most suitable for complex terrain and larger distances? Add suggestions for other 21 model(s) that would be suitable in this case.

We acknowledge that the Kapuni site is somewhat unique in this respect. As requested, we have added a paragraph at the end of section 3.4 discussing the advantages that the Kapuni site offers with regards to atmospheric transport modelling and have listed several other models that are applicable at larger distance scales and with more complex terrain and that would be more appropriate for regional-scale studies. The added text reads (p. 19 lines 23-30):

The Kapuni site has several advantages that simplify the modelling component of this method: the terrain is flat, and there are trees conveniently located close to the CO2ff sources. With larger distance scales and/or more complex terrain, WindTrax might not be an appropriate choice of model. Alternative atmospheric transport models that are applicable to larger

- 1 distances (hundreds of kilometres and/or regional scales) and more complicated geographic
- 2 features include CALPUFF (Scire et al., 2000), WRF-CHEM (Grell et al., 2005), and
- 3 AERMOD (Cimorelli et al., 2005). While these models would need to be tested in the context
- 4 of our method, the same general principles would apply.
- 5 References added:
- 6 Cimorelli, A. J., Perry, S. G., Venkatram, A., Weil, J. C., Paine, R. J., Wilson, R. B., Lee, R.
- 7 F., Peters, W. D., and Brode, R. W.: AERMOD: A Dispersion Model for Industrial Source
- 8 Applications. Part I: General Model Formulation and Boundary Layer Characterization, J.
- 9 Appl. Meteor., 44, 682–693, doi: http://dx.doi.org/10.1175/JAM2227.1, 2005.
- 10 Grell, G. A., Peckham, S. E., Schmitz, R., McKeen, S. A., Frost, G., Skamarock, W. C., and
- 11 Eder, B.: Fully coupled "online" chemistry within the WRF model, Atmos. Environ., 39,
- 12 6957-6975, 2000.
- 13 Scire, J. S., Strimaitis, D. G., and Yamartino, R. J.: A user's guide for the CALPUFF
- 14 dispersion model, Earth Tech, Inc, Concord, Massachusetts, USA, 2000.
- 15
- 16 Specific comments:
- 17 Check the table captions. Information is missing (e.g. Table 2 column 4 not explained).
- All table captions have been expanded and the requested information has been added to theTable 2 caption:
- 20 Table 2. Eight year Modelled mean CO2ff and standard deviation (SD) of eight hypothetical
- 21 sensors for simulated over the eight years 2004-2011 with of constant emissions.
- 22 Measurement uncertainty (MU) of 1.0ppm is added to the standard deviation in the fourth
- 23 column. simulation and Columns 5-10 show the detection limits calculated at the two-sigma
- 24 (95%) and one-sigma (68%) confidence level (CL) for samples representing an average of
- 25 one, two, or four years. Measurement uncertainty (MU) of 1.0ppm is added in quadrature to
- 26 *the standard deviation of modelled CO2ff before limits are calculated.*
- 27
- 28 Check figure captions. The reader should understand what those figures represent without
reading the text.

- 1 Figure captions have been expanded where possible.
- 2 Figure 2: name the two panels a) and b) and refer to them in text accordingly. Expand the
- 3 caption.
- 4 Done.
- 5 Figure 4: Same as for Fig. 2.
- 6 Done.
- 7 Page 5, line 18: 2008 should be 2007.
- 8 Thank you for catching this error.
- 9

Page 17, lines 26-28: "Indeed, looking at the results in Fig. 4, there is no significant decline at the chestnut tree in 2007; there is a small decline in CO2ff at the pine tree but it is too small to conclude that emissions have changed. "As I estimate from the figure, the observed value is smaller in 2007 than in 2006 at the chestnut tree by 1.3 ppm, and by 0.3 ppm at the pine tree. Isn't the former significant? The referee is correct that the 2007 observed value at the chestnut tree is lower than the previous year by 1.3 ppm. However, the meaning of the words "significant decline" in this context refers to the decline relative to the long-term mean (which is 2.1 ppm for the chestnut 18 tree). With respect to the long-term mean, the decline in 2007 is only 0.4 ppm (or 19% of the 19 mean), which is not enough to declare it statistically significant. We have added specific 20 numbers to the text to clarify this point (p. 17 lines 20-25):

For a one-year observation from the pine tree, this is 18%; for the chestnut, it is 92%. The largest change in emissions in any single year at the Vector plant is in 2007, with a decline of

- 23 14% relative to the long-term mean, still below the detection limit. Indeed, looking at the
- 24 results in Fig. 4, there is no significant the decline (0.4ppm, or 19% of the mean) at the chestnut tree in 2007 is not significant; there is also a small decline (0.7ppm, or 13% of the mean) in CO2ff at the pine tree but it is again too small to conclude that emissions have

- 27 changed.
- 28
- 29 I recommend using the same scale for the two graphs.

- 2
- 3

**4 Comments from Anonymous Referee #2:**

Comments: I have questions about the meteorology used for the modeling. You compare the 6 limited data set at Kapuni, close to the sampling site, with the much more complete set at 7 Hawera, 20 km southwest of the sampling site. You state that the correlations between wind 8 speed and wind direction between the two sites are consistent enough to warrant using the 9 complete Hawera data set, as shown in a direction comparison for limited dates during 10 August-October 2012 (Fig. S2). But is the limited period in 2012 adequate for evaluating whether Hawera data are appropriate for modelling wind transport at the Kapuni site? 11 12 Moreover, Figures 2 and S1 show that the wind speed at Hawera (6-7 m/s) averages on the 13 order of twice that at Kapuni (2-3 m/s). Have you done any sensitivity calculations to see how 14 this difference in wind speed affects the modeling? The wind directions seem to be fairly 15 consistent at the two sites.

See response to first comment from referee #1.

*p.1, line 25: change "lowers" to "is reduced"*

Done.

p.2, line 9: rearrange "reduction targets are commonly agreed as" to "commonly agreed
upon reduction targets are"

Done.

- 23 p. 3, lines 16-17: You mention here and again later "the [photosynthesis] process faithfully
- 24 recording the 14C content in new plant material", but you only reference the work showing
- 25 this significantly after the mention on p. 11. It might help the reader to have this discussion
- 26 *earlier, since it is critical to the method.*
- 27 We believe the referee is referring to the references and full description on p. 6 (rather than p.
- 28 11). The text in the introduction has been rearranged, but we have chosen to leave the detailed
- 29 discussion and references for section 2.3 (p. 3 lines 15-23):

Plant material can be used as a proxy for atmospheric  $CO_2$ ff because plants assimilate 2 carbon from the atmosphere during photosynthesis, in the process faithfully recording the  $^{14}C$ content in new plant material. The radiocarbon content in tree rings has been well 3 4 established as a tracer for fossil CO2 emissions (Suess, 1955; Tans et al., 1979; Djuricin et 5 al., 2012; Rakowski et al., 2013) and as a method to detect leaks from CO2 geosequestration (Donders et al., 2013). Tree rings represent an integrated average of daytime  $CO_2$ 6 atmospheric mole fractions and 14C content over the tree's annual growth period, and can be 7 8 independently dated using dendrochronology methods. This allows for a retroactive analysis 9 of  $CO_{2}$ ff mole fractions over many years, including any trends in emissions that occurred 10 during the life of the tree. The radiocarbon content in tree rings has been well established as a tracer for fossil CO2 emissions (Suess, 1955; Tans et al., 1979; Djuricin et al., 2012; 11 Rakowski et al., 2013) and as a method to detect leaks from CO2 geosequestration (Donders 12 et al., 2013). 13 14 15 p. 5, line 18: "2008" should probably be "2007". 16 Thank you for noticing this error. Figures: In general, increase font sizes for labels. Label panels within figures "a", "b", "c" 17 18 to make it easier to refer to them in the text. 19 Done. 20 Figure 1: Can you add a large-scale location map locating Taranaki in New Zealand, as well 21 as Hawera and Mount Taranaki? Add a label for Kapuni stream. 22 Done. 23 Figure 2: Font sizes. Label the legend (m/s). 24 Done. 25 Figure 4: The bottom axis of the top panel is missing. Increase the font size of the axis tick 26 labels in all panels. The dates don't line up between the top two panels and the bottom panel. 27 Increase all font sizes for the bottom panel. You use a subscript for CO2 in the bottom panel, 28 but not in the top two. In the caption: "Dotted and dashed lines show modeled and observed 29 six-year means, respectively."

Done.

- 1 Figure 5: What do the different colors for the circles indicate? The legend only shows the
- 2 *purple color.*
- 3 The colours indicated the model sensor and were redundant (because the x-axis indicates the
- 4 sensor as well). We have changed the colour of all of the circles to a single colour to avoid
- 5 confusion.

**Detecting long-term changes in point source fossil CO2 emissions with tree ring archives**

**4 E. D. Keller1, J. C. Turnbull1,2 and M. W. Norris1**

[1]{National Isotope Centre, GNS Science, Lower Hutt, New Zealand}

[2]{CIRES, University of Colorado at Boulder, CO, USA}

Correspondence to: E. D. Keller (l.keller@gns.cri.nz)

**9 Abstract**

[revised manuscript text omitted]

One of the biggest challenges of atmospheric observations of CO2ff is distinguishing the 4 fossil component from the considerable background level of  $CO_2$  that occurs naturally in the 5 atmosphere, currently about 400 parts per million (ppm; Mauna Loa observation record, 6 http://www.esrl.noaa.gov/gmd/ccgg/trends/index.html, last access: 13 May 2015). In addition, 7 there are large diurnally and seasonally varying CO2 fluxes from the biosphere, which may 8 result in changes in CO2 mole fraction of tens of ppm within a single day at near-surface sites (e.g. Miles et al., 2012). This problem can be avoided by using the 14C isotopic content as a 9 tracer for CO2ff. CO2ff contains no 14C: the half-life of 14C is 5,730 years (Karlen et al., 10 1968), and all of the 14C has decayed away from fossil fuels. Other sources of CO2 have 11 roughly the same 14C content as the atmosphere. By measuring the 14C content of CO2 or a 12 proxy for CO2, we can calculate the portion of observed CO2 
[revised manuscript text omitted]

(1)

where  $C_{\rm ff}$  is CO2ff,  $C_{\rm obs}$  is the CO2 mole fraction in the observed sample,  $\Delta_{\rm obs}$  and  $\Delta_{\rm bg}$  are the 2  $\Delta^{14}$ C of the observed sample and background sample, respectively.  $\Delta_{\rm ff}$  is the  $\Delta^{14}$ C of CO2ff, 3 and is assigned to be -1000%.  $\Delta_{bg}$  is from the summer season average from the long-term 4 Wellington 14CO2 record at Baring Head. Comparison of this record with tree rings collected 5 3 km upwind of our source showed no difference from the Wellington record.  $\beta$  is a small 6 correction to account for the fact that the  $\Delta^{14}$ C of CO2 from other sources may be slightly 7 8 different from that of the atmosphere; in our case we set  $\beta$  to zero since the proximity to the 9 coast and consistent